# Effectiveness of *Cymbopogon citratu*s Oil Encapsulated in Chitosan on *Colletotrichum gloeosporioides* Isolated from *Capsicum annuum*

**DOI:** 10.3390/molecules25194447

**Published:** 2020-09-28

**Authors:** Adriana Patricia Tofiño-Rivera, Glorismar Castro-Amaris, Fánor Casierra-Posada

**Affiliations:** 1Corporación Colombiana de Investigación Agropecuaria-AGROSAVIA, Research Center Motilonia, Km 5 vía a Becerril, Agustín Codazzi 202050, Colombia; atofino@agrosavia.co; 2Faculty of Health, Microbiology Program, Universidad Popular del Cesar, Research Group in Parasitology—Millennium Agroecology, Valledupar 200002, Colombia; gcastroa@unicesar.edu.co; 3Faculty of Agricultural Sciences, Research Group in Plant Ecophysiology, Universidad Pedagógica y Tecnológica de Colombia (UPTC), Tunja 150001, Colombia

**Keywords:** encapsulation, lemongrass, anthracnose, citral, sweet pepper

## Abstract

One of the principal etiological agents associated with losses in horticultural crops is the fungus *Colletotrichum* sp. This study aimed to evaluate the in vitro effectiveness of the essential oil (EO) from *Cymbopogon citratus* in chitosan supports for the control of *Colletotrichum gloeosporioides* isolated from sweet pepper plants. Methods: The extraction and phytochemical analysis of the EO of *C. citratus* were performed along with its encapsulation in chitosan-agar in order to compare it with other techniques and determine its effect on *C. gloeosporioides*. Results: The EO from the citral chemotype (58%) encapsulated in the chitosan-agar, with an 83% encapsulation efficiency in mass percentage, resulted in the total inhibition of mycelial growth at a minimum inhibitory concentration of 1370 ppm. This concentration was effective in controlling the disease under greenhouse conditions. The effectivity of the capsules containing EO was superior to that of other controls using EO evaluated in vitro. The capsules demonstrated an effective period of 51 days, with an additional 30 days of effectiveness after a reinfection cycle, thus providing similar results to the control with *Trichoderma* sp. Conclusions: Chitosan capsules present a promising strategy in the use of *C. citratus* EO on *C. gloeosporioides*, and they are highly effective and stable under in vitro and field conditions

## 1. Introduction

Horticulture is an important economic sector in Colombia and has promising exportation prospects; however, most horticultural products face restrictions in countries such as Canada, the United States, and Japan, among others. These countries require that crops be grown using clean production systems, either organic agriculture or sustainable agriculture with minimal chemical pesticide use [1]. It is estimated that *Colletotrichum gloeosporioides* (*Glomerellaceae*) caused losses in 48% of the vegetable crops grown in Colombia. Due to the excessive use of agrochemicals needed to control its spread, crops are unable to be exported, especially those grown by small farmers [2,3]. *Colletotrichum* is a large genus of fungi that includes a significant number of major fungal pathogens that cause diseases in various tropical and subtropical vegetables. It was listed as the eighth most important group of phytopathogenic fungi in the world based on perceived scientific and economic significance [4].

New strategies for integrated crop management, which are more efficient and provide greater coverage, must be developed. Natural pesticides that have low toxicity, high structural diversity, and low persistence can be implemented by utilizing different species whose essential oils (EOs) are found to act as potential biopesticides, allowing for the possibility of greater horticultural exportation [5]. Antifungal activity against *Colletotrichum* sp. (*Glomerellaceae*) has been demonstrated with the use of lemongrass EO (*Cymbopogon citratus*—*Poaceae*) and its major components, which produced an antifungal index value of 97.7% [6].

Limitations on the direct application of EOs in crops are related to their physicochemical properties, which include low water solubility, ease of oxidation, and rapid volatility [7]. Rapid volatility substantially reduces the duration of the effect that biocides have. This makes it necessary to develop appropriate application techniques to maximize the effectiveness and the potential of EO based formulations [8].

The agroecological conditions in the hot and dry Caribbean region of Colombia demand innovative strategies for the use of EOs so that they are stable and retain their effectiveness over time. EOs that do not lose biocidal capacity are needed so that they can be used in the production systems for crops destined for export. Additionally, the active components of EOs encapsulated in polymeric matrices such as chitosan provide formulations that can be controlled-release, biodegradable, sustainable, and have a low environmental impact [9]. Chitosan is the major component of crustacean shells that are generally discarded by the fishing industry [10]. Controlled-release systems are widely used due to their versatility and low cost. They are utilized since they act as a barrier against moisture, oxidation, and solute migration as well as for their mechanical properties, such as tensile strength and tear resistance [11,12]. These systems provide protection to the active components of the EO against various abiotic factors such as heat, pH, oxygen, and light, which can alter the effectiveness of the active ingredient [13]. They also allow a controlled release [8,9] which limits EO losses and avoids reactions with other compounds present in the environment, such as oxidation due to the presence of light and oxygen. The in vitro effectiveness of the *Cymbopogon citratus* (*Poaceae*) EO utilizing a chitosan-agar polymer matrix for the control of *Colletotrichum gloeosporioides* (Glomerellaceae) isolated from *Capsicum annuum* (Solanaceae) was evaluated in this study.

## 2. Results

### 2.1. Isolation, Identification, Reactivation, and Virulence of C. gloeosporioides

From samples of sweet pepper cv Topito plants (*Capsicum annuum*—Solanaceae) with anthracnose symptoms, the pathogen *Colletotrichum gloeosporioides* was isolated. It grew into a characteristic colony with gray-white mycelium and straight, cylindrical, hyaline, and obtuse conidia in the apex, coinciding with the description reported by Ogawa [14]. This was also confirmed through molecular analysis, as shown in Figure 1. Additionally, the reactivation of the phytopathogen in the “agar-sweet pepper” was achieved with a count higher than the inoculated 1 × 10^6^ Colony Forming Units. This concentration was evaluated in relation to its virulence and a 100% mortality rate was obtained in inoculated seedlings. The seedlings showed wilting and mycelium growth in the peppers and leaves when incubated in a wet chamber.

### 2.2. Essential Oil Extraction and Characterization

The major components found in the *Cymbopogon citratus* sample were Mircene (14.0%), Neral (22.1%), and Geranial (32.9%) (Table 1). The chemotype for this sample corresponds to Citral (55.0%) (Geranial + Neral) according to Alves [15]. Some reports mention that the biocidal activity of the EO depends on the content of sesquiterpenes and monoterpenes, which vary according to the chemotype, crop conditions, and environmental conditions prior to harvest. Citral has been identified as one of the major components in both *Cymbopogon citratus* (Poaceae) and *Lippia alba* (Verbenaceae) and is part of the group of volatile components associated with biological activity along with myrcene, eugenol, and linalool [15].

GC-FID analysis showed that the lemongrass oil used was dominated by oxygenated monoterpenes. Geranial and neral are the isomers of citral, with geranial representing the trans-isomer and neral representing the cis isomer. The retention times of the resulting three main peaks were 2.46, 13.39, and 13.96 min for myrcene, neral, and geranial, respectively. The first peak represented myrcene, the second peak represented neral, and the third peak represented geranial (Figure 2).

### 2.3. Preparation of Chitosan-EO Capsules of C. citratus

A total of 668 chitosan capsules containing EO of *C. citratus,* and 668 chitosan capsules with corn oil were prepared with an average weight of 0.07 g. The capsules were transparent and had a smooth surface, a semi-rigid consistency, and a flat surface on one side with a convex shape on the other side. Each capsule was approximately 4 mm × 1.8 mm in size and had a weight of 0.07 ± 0.007 g (Available online as Appendix A: 3—photographic record of capsule preparation and in vitro evaluation (fungi control)).

### 2.4. In Vitro Release

The in vitro EO release shown in Figure 3 represents the release profile of the optimized batch (in triplicate) of capsules, which were found to be almost identical to each other and exhibited the minimum burst effect. All the capsules (Chitosan-Agar-EO) have shown the continuous release of their content (9.7 ± 1.02%) during the 120 h of the study period, with checks every 12 h. Following the trend line, the capsule would release 100% of the EO in 1233 h, which is equivalent to 51.4 days in conditions similar to those in vitro.

### 2.5. Percentage of Encapsulated Active Ingredient

(1)Content a.i (%)=(33.32 46.79 )×100=71.21%

The preparation of the capsules resulted in the use of 33.32 g of EO, 36.75 g of chitosan, and 1.88 g of Agar to obtain 668 capsules with a total weight of 46.7 g. This indicates that the 3:1 ratio (Chitosan-Agar-EO) used as an encapsulant allowed the final content of the active ingredient in the capsule to be 71.21%. An increase in the polymer content of the formulation increases the retention capacity of the active ingredient.

### 2.6. Mass Encapsulation Efficiency

According to the analytical chemistry results, the total oil retained corresponded to 28.4 g, while the surface oil retained was 4.8 g. This yielded an 83% mass encapsulation efficiency as a result of the 33.32 g of EO added to the initial mixture for the preparation of the capsules, of which 28.4 g was maintained.
(2)$$EE\;\left(\%\right)=\frac{28.4g-4.8g}{28.4g}\times \;100=83\%$$

### 2.7. Chitosan Antifungal Activity Against C. gloeosporioides

In order to assess whether chitosan alone had some antifungal activity, its activity on the pathogen was determined. According to the results obtained, no significant difference was identified between the chitosan treatment alone and the control treatment with agar-PDA following 24, 48, and 72 h of incubation. This indicates that this polymer matrix does not inhibit the growth of *C. gloeosporioides* efficiently. A mycelium radius of 0.25 cm was present around the chitosan capsules, giving an equivalent inhibition of mycelial growth of only 3.1%.

### 2.8. Antifungal Activity of Chitosan-Agar-EO Capsules of C. citratus Against C. gloeosporioides

Every chitosan-agar-EO treatment of *C. citratus* showed significant differences in the percentage of inhibition of the mycelial growth of *C. gloeosporioides* after 72 h of incubation. The control treatments using corn oil resulted in no inhibition of the mycelial growth of the pathogen. However, treatments with 913, 1056, and 1156 ppm of *C. citratus* EO, corresponding to 20 capsules in each of the three systems, were less efficient in controlling the pathogen since they showed inhibitions below 80% with 43%, 49.7%, and 53.9% inhibition of mycelial growth, respectively. The treatment with 1370 ppm in chitosan-agar-EO capsules of *C. citratus* showed the minimum inhibitory concentration in vitro with 100% growth inhibition of *C. gloeosporioides*.

### 2.9. Comparison of Antifungal Activity Effectiveness Using Different Application Techniques of EO against C. gloeosporioides

When comparing the different in vitro techniques for the use of *C. citratus* oil against *C. gloeosporioides*, it was observed that chitosan-agar-EO capsules had a 100% inhibition on the growth of the pathogen. Significant differences were found between 5 and 30 days after starting treatment, comparable only with the positive control corresponding to the *Trichoderma* antagonism technique (with prolonged effect) (Figure 4). The other techniques used, such as the poisoned food and wet chamber technique, reached values of inhibition of mycelial growth of 88.5% and 77.5%, respectively, after five days. However, their effectiveness decreased until they became ineffective 30 days after starting treatment (Table 2).

### 2.10. Evaluation of the Resistance of (Chitosan-Agar-EO) Capsules under Environmental Conditions

During the tests for resistance against environmental conditions, there was no weight loss in capsules for the first 15 days, and they maintained their initial average weight of 0.07 ± 0.007 g. Subsequently, after 30 days, there was a weight reduction of 50% with respect to initial size. After 45 days, the weight of the capsules was 10% of their initial weight (Figure 5 and Figure 6). Likewise, it was found that during the first 15 days, the capsules were firm, but between 15 and 36 days, they became soft, and after 45 days, their consistency was glutinous.

The storage stability of the capsules was tested by keeping them refrigerated at 8 °C for 15 months. There was no loss of physical characteristics such as color change or loss of firmness. The weight of each capsule had a slight decrease of 0.0035 g on average, which is equivalent to 5% based on an initial weight of 0.07 g.

### 2.11. Antifungal Activity of (Chitosan-Agar-EO) Capsules of C. citratus against C. gloeosporioides in Topito Pepper Plants: Measurement of the MIC Test in a Greenhouse.

During testing of the antifungal activity of the (Chitosan-Agar-EO) capsules in greenhouse conditions (Figure 7), approximately 10 to 51 days after inoculation, all of the control treatments showed signs of infection. The plants treated with 20 capsules showed mild signs of infection by *C. gloeosporioides* in the three plants. Plants treated with 30 and 40 EO capsules did not show signs of infection. A minimum inhibitory concentration was found with the use of 30 capsules with 255 µL of the EO from *Cymbopogon citratus* (Figure 8 and Figure 9).

An incidence of 66% of anthracnose was found as 12 of the 18 inoculated plants showed signs of disease. Of these 12 plants, nine corresponded to the control treatment with corn capsules and three to the treatment with 20 EO capsules of *C. citratus*. Therefore, the latter did not present symptoms. The percentage and degree of severity of anthracnose in Topito peppers with treatments of EO capsules of *C. citratus* and Corn Oil controls is shown in Table 3.

Treatments with 30 and 40 EO capsules for the inhibition of *C. gloeosporioides* showed significant differences (*p* ˂ 0.05) in the % MIC with respect to using 20 EO capsules for 10 days as well as against controls with corn oil. Treatment with corn oil did not reveal any significant differences between them.

Controls with 20, 30, and 40 capsules of corn oil presented a percentage of damage of 84.6%, 90.2%, and 94.6%, respectively. Treatment with 20 EO capsules showed slight signs of infection with a damage percentage of 32%. Treatments with 30 and 40 EO capsules did not show any signs of damage.

## 3. Discussion

Samples taken from sweet pepper cv Topito plants with symptoms of *C. gloeosporioides* and with 100% mortality of inoculated seedlings in vitro confirmed that the pathogen is present in the dry Caribbean region of Colombia. As the causative agent of anthracnose in *Capsicum annuum*, the pathogen presents a serious threat to its horticultural production in the tropical and subtropical regions where this vegetable is grown, causing necrotic lesions and limiting its quality and possible commercialization. Severe pathogen attacks result in production losses that are greater than 80% [4], demonstrating the need for the development of strategies for the sustainable control of the phytopathogen *C. gloeosporioides.* Additionally, it is important that the tests developed in vitro are carried out using highly virulent strains as was the case of this research study. Likewise, isolation from the area being studied helps to increase the margin of reliability in the results obtained with the use of the EO application technique.

*C. citratus* EO had a citral component of 55% which was slightly below the levels referred to in other studies where concentrations of nearly 60% of this bioactive mixture (neral + geranial [C_10_H_16_O]) were present [16]. The composition of the EO varies in relation to the genetic component of the plant, its cultivation conditions, harvest time, the harvested part of the plant, the management of the plant material, and the extraction methods, among other factors. Researchers [6,17], evaluated the antimicrobial effect of the individual citral and geranial components present in *Cymbopogon* sp on the horticultural pathogens *Rhizoctonia solani* (*Ceratobasidiaceae*), *Fusarium oxysporum* (*Nectriaceae*), and *Sclerotium rolfsii* (*Atheliaceae*). A variant of the poisoned food technique was used to verify the efficiency of these molecules and its selectivity according to the phytopathogen evaluation of the biocontroller. Citral was more effective in the control of *Fusarium oxysporum* (*Nectriaceae*), reaching 85% of the inhibition of radial mycelial growth with the addition of 0.4 µL mL^−1^, while geranial was much more effective in the control of *Sclerotium rolfsii,* reaching 85% growth inhibition when 2 µL mL^−1^ was used. This level of inhibition of the individual molecules of the *C. citratus* EO as citral could suggest a possible synergistic effect of the 24 components identified in the EO on the growth control of *C. gloesporioides* since 100% inhibition of mycelial growth was achieved in this study.

Consequently, the use of EO in chitosan capsules covered with a pure agar-agar membrane was optimized, representing an easier method of implementation coupled with greater effectivity during microencapsulation [10]. This is beneficial since the loss of EO because of reactions with other compounds in the medium and oxidation reactions due to light or oxygen is avoided. The observed stability of the capsules elaborated in the present study is associated with several benefits. The first is the favorable structure, solubility, and gelation of the agar-agar, which is a complex mixture of polysaccharides composed of agarose and agaropectin polymers, which make it an excellent encapsulant due to their solubility properties. This material is tolerant to rainfall and maintains its stability at temperatures below 50 °C. The second benefit is that chitosan is a primarily hydrophobic linear copolymer with a rigid structure that is insoluble in water or in organic solvents and that has low solubility in contact with basic substances. Other characteristics that make it water resistant are the distribution of acetyl groups along the macromolecular chain, the concentration of the polysaccharide, and the ionic strength [18]. The third benefit is that under the edaphoclimatic conditions of the vegetable producing areas in the dry Caribbean region of Colombia near Serranía del Perijá and the Sierra Nevada de Santa Marta the possibility of accelerated deterioration of the capsules is reduced due to the predominance of slightly acidic and neutral soils (Vertisols, Entisols, Aridisols, and Fluvents). Additionally, due to the short and well-concentrated periods of rainfall during four months of the year, compared to other regions of Colombia, implementation is easier. This allowed the capsules to maintain their durability during periods of minimal precipitation in arid zones [19].

As evidenced during the resistance test to environmental conditions, the capsules maintained their stability during the 45 days of evaluation. There is evidence of a prolonged effect during the greenhouse test of antifungal activity, in which a MIC was identified corresponding to a concentration of 255 µL in 30 capsules and an inhibitory range of 100% for the development of anthracnose by *Colletotrichum gloeosporioides* during the 12 days after inoculation. Other studies have reported the appearance of the first symptoms of infection up to 51 days after inoculation [20]. The capsules have demonstrated their extended effect in controlling the release of the remaining inoculum.

In addition, Mishra and Dubey [21] found *Cymbopogon* oil to be more effective than carbendozim in pathogen control under in vitro conditions. Another study reported the application of chitosan as a crop management strategy to efficiently induce biotic and drought stress resistance in lentil plants in semi-arid regions [22]. On the other hand, the healthiness of bean plants grown in soil treated with chitosan and an increased quantity of microorganism colonies was significantly better compared to the control. The populations of antagonistic microorganisms formed in the soil in these treatments may have limited the growth of pathogenic fungus [23]. The results of ecological studies and an increase in yield involving EO and chitosan suggested that these molecules provide health benefits to the soil [22,23]. The chitosan used in this study did not show antifungal activity, which differed from the results obtained by Younes and Rinaudo [10], who used chitosan to control fungi in several food categories. Other studies indicated that the molecular weight and the degree of deacetylation affect the biological activity of chitosan. The inhibition of phytopathogens is partial in both products with higher degrees of chitooligosaccharides deacetylation or low deacetylation-Water-soluble chitosan products at the highest evaluated concentrations [18].

The chitosan used to make the capsules showed no biocontrol effect on the pathogen evaluated. The non-significant inhibition by chitosan when used individually against *C. gloeosporioides* as well as its antimicrobial activity referred to in this study depends on both the surface characteristics of the pathogen cells and the characteristics of the chitosan. Depending on the process and the conditions for obtaining chitosan, it acquires a greater or lesser degree of deacetylation (DD) and positive load density (PLD). With a DD of 97.5%, PLD increases, which also increases antimicrobial activity [24]. The chitosan selected for this study had a lower degree of deacetylation, so it did not act as an antimicrobial agent but only as an encapsulating agent. This facilitated the analysis of the results of the antimicrobial activity of the EO obtained from *Cymbopogon citratus*. Other studies conducted by Gutiérrez-Martínez [25] report an inhibition of 80% to 100% on isolates of *Colletotrichum* sp using the polymer at 1%, molecular weight of 1.74 × 10^4^, and a deacetylation of 75–85%.

The antifungal activity of (Chitosan-Agar-EO) capsules against *C. gloeosporioides* showed that treatment with the EO of *C. citratus* at a concentration of 1370 ppm had a 100% inhibitory effect on the pathogen. Some studies determined the effectiveness of the essential oil of *C. citratus* with respect to others from *Lippia hadides* Cham (*Verbenaceae*) and *Ocimum gratissimum* L. (*Lamiaceae*) The essential oil from *C*. *citratus* yielded the highest mycelial growth inhibition at all concentrations (1, 3, 5, and 7 µL mL^−1^) [26]. Other studies found a high level of EO control against *C. gloesporioides* at different concentrations. This depended on the plant and concentration, which varied between 0.80% (*Melaleuca alternifolia*), 3.20%, (*Eucalyptus globulus*), and 6.25% (*Citrus limonum, Rutaceae; Cymbopogon citratus Poaceae; Syzygium aromaticum, Myrtaceae; Cinnamomum zeylanicum, Lauraceae* and *Azadirachta indica, Meliaceae*) [27]. Different results were reported by Marcondes [28] on the *Colletotrichum gloesporiodes* control by *Cymbopogun citratus, Poaceae*; *Allium sativum Amaryllidaceae*; *Rosmarinus officinalis*, *Lamiaceae*; *Origanum vulgare*, *Lamiaceae*; *Salvia officinalis*, *Lamiaceae*; *Piper nigrum*, *Piperaceae* and *Caryophillus aromaticus, Myrtaceae* aqueous extracts. The best results were found by inhibiting the sporulation and the mycelial growth of the pathogen, demonstrating that the EO is a better method of maximizing the biocontrol potential of *Cymbopogon citratus*.

Similarly, the antimicrobial activity of other essential oils in chitosan films of different molecular weights on bacteria was evaluated. Hernández-Ochoa [29] found 100% mycelial growth inhibition using high molecular weight chitosan films with EOs from cumin (*Cuminum cyminum*) (750 ppm), cloves (*Eugenia caryophyllata*) (500 ppm), and helenium (*Inula helenium*) (500 ppm) against *Escherichia coli*, *Enterobacteriaceae; Salmonella typhimurium*, *Enterobacteriaceae; Staphylococcus aureus*, *Staphylococcaceae*; *Bacillus cereus*, *Bacillaceae* and *Listeria monocytogenes, Listeriaceae.* These films can be considered potentially useful as an active packaging material and can help prevent and control the spread of pathogenic microorganisms in food.

Chitosan capsules showed a more effective means of applying EO than using the poisoned food and wet chamber techniques. However, there is a special interest in the development of bio-inputs and the use of nanoemulsions of different EOs such as *Azadirachta indica, Meliaceae* and *Cymbopogon nardus, Poaceae*. Both were successfully tested in in vitro conditions by means of the poisoned food technique for the control of phytopathogenic fungi of economic importance in horticulture like *Rhizoctonia solani, Ceratobasidiaceae* and *Sclerotium rolfsii, Atheliaceae*. The nanoemulsion of EO of *C. nardus* demonstrated a better control over *R. solani* (ED50 13.67 mg L^−1^) [30]. At the in vitro level, the process of reinoculation of the pathogen after 30 days demonstrated 100% inhibition, confirming the effectiveness of the use of a chitosan polymer matrix combined with a solidifying agar-agar matrix. This study presents the first report on the effectiveness of the agar-agar-chitosan complex in forming an encapsulation complex for the active components of the *C. citratus* EO as well as its subsequent controlled release.

The current study also noted that the chitosan-agar encapsulation optimizes the oil’s action and yield times, by presenting 100% effectiveness in the control of *C. gloeosporioides* after the first five days of treatment. The other techniques evaluated showed an effectiveness of less than 90% MIC and failed to sustain their effect after 30 days. Additionally, it should be mentioned that EO differentially impacts fungal cells depending on the miscibility in the pathogen’s cell membrane. Here, it interacts with proteins, causing changes in hyphae and plasma membrane and causing alterations in their physicochemical properties (hydrophobicity, electrical conductivity, soluble protein filtration, and sugar reduction), which ultimately cause apoptosis [31]. The control of the EO of *C. citratus* over *C. gloesporioides* is efficient regardless of the technique used as the literature refers to the control of mycelial growth equal to or greater than 80% [19].

Chitosan capsules maintain their effectiveness due to the slow release of the encapsulated substance while also improving stability and solubility, which prevents the loss of EO [25]. This is the same reason why they are widely used in the food and pharmaceutical industry. This represents an economic benefit, making the use of the EO of *C. citratus* for the control of phytopathogens in vegetables more economical than the use of conventional agrochemicals. This is especially the case as chitosan is an abundant byproduct obtained from the fishing industry [10,32]. The use of the EO of *C. citratus* and *Trichoderma* sp. in integrated disease management systems was found to inhibit the growth of the antagonist by 100% seven days after inoculation using the poisoned food technique, as shown by Melo [3]. In the case of chitosan-EO capsules, inoculations of *Trichoderma* sp. in sweet pepper plants should be done 60 days after the addition of the EO capsules to avoid the mentioned fungitoxic effect. A period of nine additional days will be necessary to limit the effect of volatile compounds resulting from the encapsulation of EO in chitosan. Another alternative for biological control in the dry Caribbean could be using pathogen control with chitosan-EO capsules of *C. citratus* during the time of low rainfall and controlling with *Trichoderma* in times of heavier rainfall. This compromise is beneficial since *Trichoderma* sp. requires certain levels of environmental humidity to favor spore germination and biomass production.

Greater use of EO in agriculture for the integrated management of fungal diseases caused by *Colletotrichum* sp. in short-cycle vegetable crops could be made. This would allow fewer applications of fungicides since the stability of the capsules is greater than 40 days. Additionally, successive applications could be avoided since the reinfection process in the culture typically occurs within 30 days. These aspects make these capsules highly competitive against other unsustainable alternatives for the management of this phytopathogen. These advantages confirm the importance of linking new strategies to the integrated management of diseases in the fruit and vegetable sector of the dry Caribbean sub-region of Colombia. Regarding the use of EO, its potential as a phytopathogen controller has not yet been exploited in this sector [33], although it presents a good alternative given its effectiveness and stability against other techniques when it is encapsulated in chitosan-agar supports.

Although *Trichoderma* sp. can be an efficient way to control *Colletotrichum gloesporioides* [34], its effectiveness, like that of other bio-inputs such as *Beauveria bassiana*, depends on environmental conditions. In especially arid areas, such as the Colombian dry Caribbean, high radiation and low relative humidity can limit the sporulation of biocontrollers [19,35]. Therefore, it is especially important to continue field evaluations of the chitosan capsules proposed in this study and verify whether the use of a biopolymer with fungistatic activity can act synergistically with EO. Future studies should include not only the effect on crop health, but also the effect on promoting plant development and pepper yield, as has been evidenced in other horticultural crops [22]. Likewise, the results of this study suggest the relevance of continuing with studies for the standardized production of *C. citratus* oil in chitosan-agar capsules and testing both in greenhouses and in open fields. The capsules’ cytotoxicity, phytotoxicity, socioeconomic impact, and environmental impact in the management of anthracnose associated with *C. gloeosporioides* in sweet pepper should be studied further.

## 4. Materials and Methods

### 4.1. Pathogen Selection

The study was conducted utilizing the facilities of the Universidad Popular del Cesar (UPC) located in Cesar, Colombia (10°18′ N and 73°24′ O). Sweet pepper seedlings (*Capsicum annuum*—*Solanaceae*) from the traditional cultivar cv. Topito were utilized for the experiments. It should be noted that this pepper is a local variety and is sweet as opposed to hot like the chili pepper variety (*Capsicum chinense*—*Solanaceae*) of the same name. The plants were between 2 and 6 months old and were planted at the UPC experimental farm in Valledupar, Colombia. The farm was located at an altitude of 169 m above sea level, with an average annual temperature of 28.4 ± 5.4 °C, an average annual relative humidity of 65 ± 9%, and 961 mm of average annual precipitation. The pathogen *Colletotrichum gloeosporioides* (*Glomerellaceae*) was isolated from the leaves and peppers of the sweet pepper plants cv. Topito following the protocol used by Melo [3]. The pathogen was isolated in potato dextrose agar (PDA) and cultured at 25 °C. A conidia suspension was then made using a concentration of 1.0 × 10^6^ CFU mL^−1^ in sterile distilled water [36]. Taxonomic keys of macro and microscopic morphology were used to identify the fungus as described by Gañán [36]. Molecular analyses were performed in the laboratories of the Corporación para Investigaciones Biológicas-CIB in Medellín, Colombia. DNA extraction was performed using the hexadecyltrimethylammonium bromide (CTAB) method. Amplification of the internal transcribed spacer region (ITS) was performed using a polymerase chain reaction (PCR) technique with universal primers TS1 and ITS4. The reactions were incubated at 95 °C for five minutes, then 30 cycles were performed at 94 °C for one minute (denaturation), at 55 °C for one minute (alignment), at 72 °C for one minute and a half (extension); then, a cycle of 72 °C for five minutes (final extension) and a final cycle for an indefinite time at 8 °C were performed.

The PCR results were sent to Macrogen (Korea) for sequencing. The forward and reverse sequences were debugged, edited, and aligned using Geneious software (Biomatters, Ltd., Auckland, New Zealand), version 9.1.5. The consensus sequences were compared with those available in the GenBank database using the BLAST application (Basic Local Alignment Search Tool) (Rockville Pike, Bethesda MD, USA) to determine the identity of the isolated pieces (Available online as Appendix A: 2—Blast in the NCBI for the isolation).

Additionally, reactivation of the pathogen was carried out using agar-sweet pepper (macerated plant material added to agar-agar) [37]. Likewise, the virulence of strains isolated from *C. gloeosporioides* was assessed using wet chamber mounts (Available online as Appendix A: 4—photographic record of in vitro evaluations of pathogen activity (Reactivation of the pathogen; Virulence of strains)).

### 4.2. Collection of Plant Material of Cymbopogon citratus (Poaceae) and Extraction of Essential Oils

The plant material was harvested manually in the rainy season from crops without the use of agrochemicals. The plants grew in moderately acidic soil with 2% organic matter in the indigenous community of Atanquez, Colombia, and were located 1200 m above sea level (10°42′15″ N, 73°17′18″ W). This community is home to the ASOPROKAN association, which is made up of agroecological producers from the indigenous community of the Kankuamos. The essential oil of *C. citratus* was extracted using steam distillation [16]. The EO was analyzed in the natural products laboratory of the Universidad de Córdoba, located in Montería, Colombia. A phytochemical evaluation was performed using gas chromatography with subsequent gas characterization/mass spectrometry (GCMS-TQ8050 NX Shimadzu, Kyoto, Japan) using a selective detector. Presumptive identification was performed using Apolar DB-5MS (60 m) and Polar DB-WAX (60 m) columns and compared with information from the NIST (National Institute of Standards and Technology) in Gaithersburg, MD, USA, and the MassLab program (Porto, Portugal).

### 4.3. Obtaining and Characterizing Chitosan Capsules

Chitosan capsules were made by using the modified chitosan-agar technique described by Jovanović [38]. Chitosan was cross-linked with agar-agar in order to encapsulate the EO. A 6 mL solution of chitosan in 2% acetic acid was prepared in an orbital shaker for 12 h. This solution was mixed with 3 mL of agar-agar and 3.6 mL of EO and agitated until homogenized. Eighty-five µL was then poured into each of the 120 microplate cavities which served as molds for the capsules and cooled for 12 h. The filling of the microplate was repeated until 668 capsules were made. The selection of the proportion of the components of the capsule was carried out from a previous test in which different proportions were tested until they reached a firm consistency after a week under ambient conditions (Available online as Appendix A: 3—photographic record of capsule preparation and in vitro evaluation (fungi control)).

The controls were obtained following the same methodology but by replacing the EO of *C. citratus* with corn oil (*Zea mays-Poaceae*)).

### 4.4. In Vitro Release

The in vitro EO releasing property of microspheres was evaluated under the maximal yield condition of encapsulation using a modified USP dissolution apparatus type I (Electrolab, Dissolution tester, EDT-08Lx, Janki Impex, Gujarat, India) assembled with a 40-mesh basket and 200 mL capacity flasks. Finally, a known volume of ethanol/phosphate buffer solution (1:9 *v*/*v*, pH 7.4) was poured into the flask, and the assembly was set at constant stirring (50 RPM) and temperature (37 ± 0.5 °C) (18). At a predetermined interval, i.e., after 0, 15, 30, 60, 120, 180, 240, 480, and 720 min, a 1 mL sample was collected followed by the replenishment with the same volume of fresh and preheated receptor medium solvent at each sampling interval. The collected samples were analyzed by UV spectrophotometer (Perkin Elmer, Waltham, MA, USA) at 223 nm after appropriate dilution. The experiment was performed in triplicate [39].

### 4.5. Percentage of Encapsulated Active Ingredient

To measure the percentage of active ingredient, the mathematical relationship described by Vila [40] with the following equation was used:(3)Content a.i (%)=(Quantity of encapsulated a.i final weight of the capsules)× 100

### 4.6. Total Oil Content

To evaluate the total oil content or ether extract, the quantitative method of total fat by acid hydrolysis described in the Official Method AOAC 922.06 [41] was used as a reference. Approximately 1 g of each sample was weighed in 50 mL beakers to which 1 mL of ethyl alcohol and 5 mL of hydrochloric acid were added. The solution was then mixed and placed in a water bath for 40 min. Then 5 mL of alcohol was added, and each sample was allowed to cool. Each treatment was transferred to 15 mL test tubes, and 4 washes with ethyl ether and petroleum ether were carried out in previously weighed glassware where the solvent was allowed to evaporate for approximately 1 h and placed into an oven at 100 °C for 1 h.

### 4.7. Content of Surface Oil

To evaluate the content of oil present on the surface of the capsules, the methodology described by Calvo [42] was followed. One g of sample was weighed on a screw cap glass bottle, and 7.5 mL of hexane was added. It was vortexed for 2 min at room temperature to extract the free oil. The mixture was decanted and filtered through filter paper and the collected powder was rinsed 3 times with 10 mL of hexane. The powder was dried in an oven at 60 °C to remove any residual solvent, and the final weight was recorded with each sample evaluated in duplicate. The encapsulation efficiency was determined according to the equation:(4)$$\mathrm{EE}=\frac{\mathrm{TO}-\mathrm{SO}}{\mathrm{TO}}\times \;100$$
where:
EE = Encapsulation efficiency as a percentage of mass.TO = Total oil retained in g.SO = Content of superficial oil in g.

### 4.8. Evaluation of the Antifungal Activity of Chitosan Against C. gloeosporioides

An adjustment was made to the methodology of the “agar diffusion assay” described by Velásquez [43]. Petri dishes were used with five chitosan capsules without EO content. The surface of the agar was inoculated with a solution of known spore concentration, and 3 triplicate control treatments were conducted every 12 h for three days.

### 4.9. Determination of the Minimum Inhibitory Concentration of Chitosan Capsules with EO for the Control C. gloeosporioides

An assay was performed for the evaluation and determination of the concentration of oil necessary for pathogen inhibition. This was based on the minimum inhibitory concentration (MIC) of 1000 ppm, which turned fungicidal at 2000 ppm as described by Mishra and Dubey [21]. The system used consisted of two glass containers. The first one was transparent and had a total volume of 129 cm^3^ minus 25 cm^3^ of PDA agar, resulting in an actual volume of 104 cm^3^. The second one was amber in color, had a total volume of 25 cm^3^, and contained the chitosan capsules. Both vessels were connected using a plastic coupling hose with a diameter of 1.2 cm. Tube lengths of 16, 28, and 50 cm were used, which had an equivalent volume of 18, 32, and 57 cm^3^, respectively (Figure 10). These variable lengths of tubing simulated the distances between the ground and the lower leaves of the plant (16 cm hose), from the soil to the middle zone (28 cm hose), and from the soil to the upper leaves (50 cm hose) of a sweet pepper plant cv. Topito. These simulations were done in order to verify whether the volatile effects of EO-agar-chitosan in capsules located in the soil around the base of the plant were able to reach different heights of the plant (Available online as Appendix A: 5—photographic record of in vitro system evaluations of pathogen activity). The length of the hoses was based on the average height of 90 plants 90 days after transplantation, resulting in three sets of volumes equivalent to 147, 161, and 186 cm^3^, respectively, according to the formula:

Volume of the system= volume of transparent container + volume of amber container + volume of tube.

The MIC of the *Cymbopogon citratus* (*Poaceae*) EO was determined by the volatile effect of sustained release systems using chitosan capsules and taking into consideration the MIC established in the preliminary test. The fungus was inoculated in PDA and exposed to nine treatments equivalent to the three concentrations of oil, 170, 255, and 340 µL (20, 30, and 40 capsules, respectively), in the three systems and using nine controls (corn oil capsules) (Table 4). Each system constituted an experimental unit and was evaluated in triplicate for a total of 54 trials. The percentage of mycelial growth inhibition (% MGI) proposed by Melo [3] was measured at 7, 15, 21, and 30 days.

### 4.10. Efficiency of Chitosan-EO Capsules in Relation to Other EO Application Techniques

The in vitro efficiency of chitosan and EO capsules was measured by comparing them with the two alternative techniques of EO application. With the PDA agar as a growth medium, these corresponded to the following treatments: PDA + 1500 ppm EO (poisoned food technique described by Melo [3]), wet chamber EO (technique described by Reyes-Chaparro [12]) in Petri dishes with agar and with one 2 cm paper disk impregnated with 60 µL of EO fastened with paper tape on the inside of the lid, chitosan capsules with EO (modified technique by Jovanović [38]), and two control treatments, one negative with PDA and one positive using the *Trichoderma harzianum* (Hypocreaceae) antagonist [44]. The treatments to determine the efficiency of chitosan were (P0) EO capsules + PDA, (P1) Poisoned Food Technique corresponding to the effect of contact using PDA + EO, (P2) Wet Chamber Technique using impregnated disks with EO, (P3) Capsule Technique (Chitosan-Agar-EO), and (P4) Antagonism Technique *Trichoderma* sp. vs *Colletotrichum gloeosporioides*.

For P0, P1, P2, and P4, a 5 mm diameter portion of PDA was inoculated on which *C. gloeosporioides* had grown. P3 utilized 1 mL of a solution of 1 × 10^6^ conidia mL^−1^ spread over the surface of the PDA. All trials were performed with 3 experimental treatments and 3 control treatments. All processes were incubated at room temperature; measurements were taken every 24 h for 7 days, with measurements of the diameter of the colonies measured at 2, 5, and 7 days. The measurements were expressed as a percentage of inhibition of mycelial growth (IMG) with the following formula: IMG (%) = [(gd − dt)/dt] × 100, where gd= is the growth diameter of the control mycelium and dt= is the diameter mycelium growth with EO. The minimum inhibitory concentration (MIC) of the oil was the lowest concentration of the oil that gave 100% inhibition in mycelial growth [3].

All in vitro experiments were performed using a completely randomized design and in triplicate. The data was statistically evaluated using an analysis of variance and the Least Significant Difference (LSD) test of Tukey. Additionally, a Probit analysis (IBM, Madrid, Spain) was performed for the calculation of the balanced scorecard, using the SPSS version 20.0 application (IBM Corporation, Armonk, NY, USA) with a *p* < 0.05 (Available online as Appendix A: 1—SPSS statistical analysis results).

### 4.11. Evaluation of the Resistance of Chitosan-Agar-EO Capsules under Environmental Conditions

The evaluation of environmental conditions was carried out using the methodology described by Males [45] while taking the environmental conditions in the Cesar region into account [46] (Table 5). Nine (Chitosan-Agar-EO) capsules were selected and placed on the soil surface of three Topito pepper plants, and this was repeated three times. This was carried out in February, July, and October in order to measure these results against the environmental conditions that presented greater values in their variables. Measurements were taken every 5 days and took into account the firmness and weight of the capsules until dry or until a weight loss of greater than 90% was recorded.

### 4.12. Antifungal Activity of Chitosan-Agar-EO Capsules of Cymbopogon citratus against Colletotrichum gloeosporioides in Topito Pepper Plants: Measurement of the MIC Test in a Greenhouse

Antifungal activity was determined by measuring the MIC in relation to the incidence and severity of infection in the plants. Taking into consideration the effect of environmental conditions on the capsules, and based on the MIC found in the laboratory, the use of 255 µL of EO distributed in 30 capsules allowed the fungus to be inoculated with a spore concentration of 1 × 10^6^ conidia mL^−1^, which had previously been used under in vitro conditions. These spores were applied on the leaves of Topito pepper plants, whose average height was 50 cm. The capsules were located on the substrate and at the base of the stems. Treatments with 170 µL (20 capsules), 255 µL (30 capsules), and 340 µL (40 capsules) of *C. citratus* EO and their corresponding controls with corn oil were performed. The experimental unit consisted of a plant planted in 5 Kg potters and capsule control using corn oil. The MIC was determined from the daily review of the experiment until the appearance of an infection was seen.

### 4.13. Incidence (%) of Anthracnose in Evaluated Plants

This variable was determined by taking into account the number of diseased peppers with respect to the number of healthy peppers in a total of 18 plants by modifying the mathematical formula described by Andrades [47].
(5)$$Incidence\left(I\right)=\frac{\mathrm{Number}\;\mathrm{of}\;\mathrm{diseased}\;\mathrm{peppers}\;\times \;100}{\mathrm{Number}\;\mathrm{of}\;\mathrm{healthy}\;\mathrm{peppers}}$$

### 4.14. Severity (%) of Anthracnose in Fruit

This aspect was evaluated in mature fruits from 18 Topito pepper plants after 120 days with approximately 37 peppers per plant using a visual scale described by Andrades [47]. According to the percentage of severity, a degree of severity for the infection is given. From 0% to 3% severity, the degree is between 0 and 1 for healthy plants, from 3% to 75% of severity, the degree is between 2 and 6 for plants with characteristic signs of the disease, and from 75% to 100% of severity, the degree is between 7 and 11 for plants with generalized wilting or death.

## 5. Conclusions

Significant differences were found between the antifungal activity of chitosan-agar-EO capsules of *C. citratus* on *C. gloeosporioides* and on the application techniques of EO in poisoned food or wet chambers. With this method, 100% MIC was maintained for up to 30 days after initial dosage, which is in contrast to a 0% MIC obtained with use of other techniques. The stability of the *C. citratus* (Chitosan-agar-EO) capsules for 45 days and a controlled effectiveness for 51 days constitute optimization in the applicability of the *C. citratus* EO. By maximizing performance and minimizing losses in the volatilization of bioactive molecules, the integrated management of diseases associated with phytopathogenic fungi of high virulence and prevalence in vegetables grown in the dry Caribbean region of Colombia can be mitigated.

## Figures and Tables

**Figure 1 molecules-25-04447-f001:**
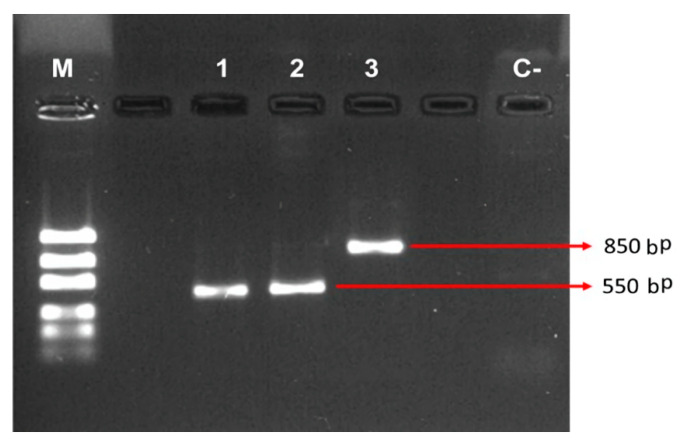
Verification of amplification of the internal transcribed spacer region (ITS) region by 1% agarose gel electrophoresis. The bp values shown are approximated. Code 2 corresponds to *Colletotrichum gloeosporioides*.

**Figure 2 molecules-25-04447-f002:**
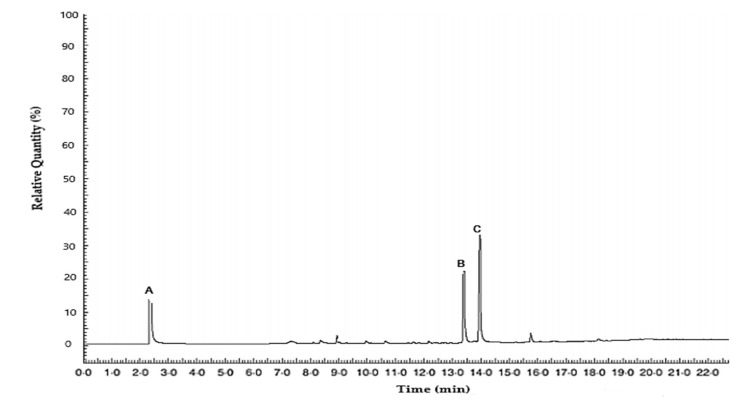
Chromatogram of the essential oil from lemongrass. The first peak (A) represents myrcene, the second peak (B) represents neral, and the third peak (C) represents geranial.

**Figure 3 molecules-25-04447-f003:**
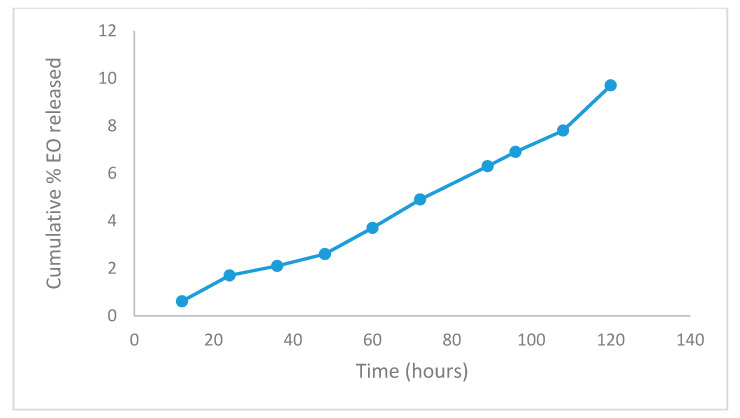
Cumulative EO percentage released from optimized capsules in a phosphate buffer (Ph 7.4).

**Figure 4 molecules-25-04447-f004:**
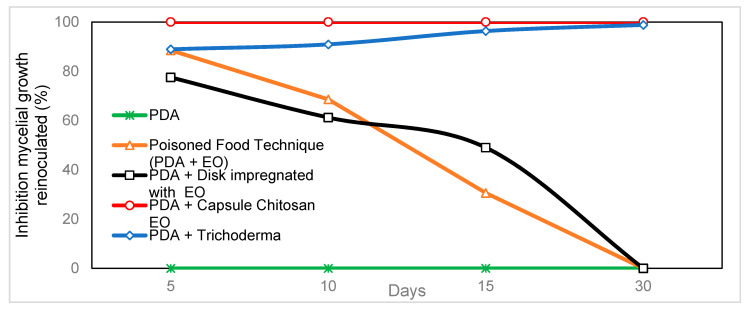
Comparison of the effectiveness of techniques using *Cymbopogon citratus* EO in the inhibition of mycelial growth of *C. gloeosporioides* up to 30 days.

**Figure 5 molecules-25-04447-f005:**
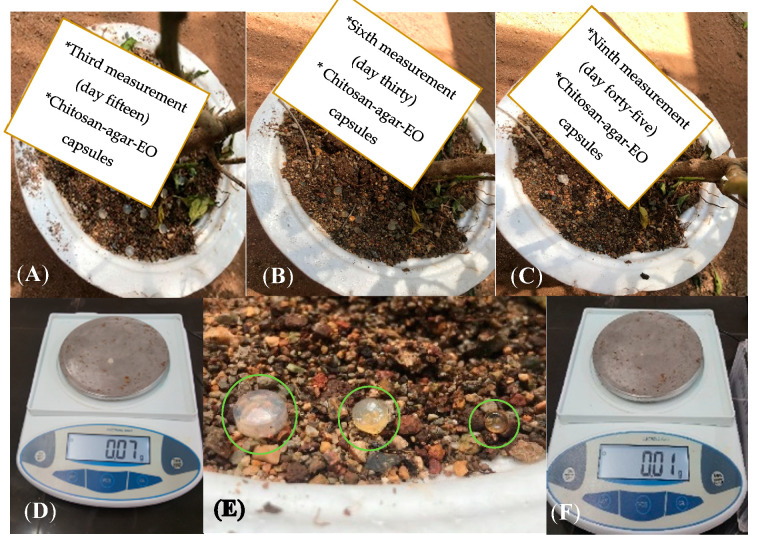
Consistency and weight measurement process during days 15, 30, and 45 of the test with a 100%, 50%, and 10% reduction in weight, respectively (**A**–**C**). Weight loss ratio of Chitosan-Agar-EO capsules from day 1 to day 45 (**D**–**F**).

**Figure 6 molecules-25-04447-f006:**
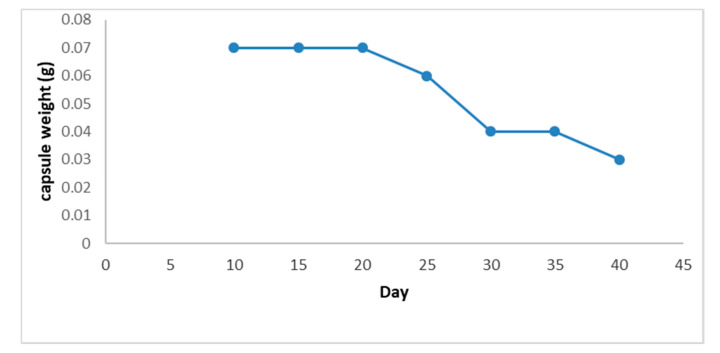
EO chitosan capsule weight stability.

**Figure 7 molecules-25-04447-f007:**
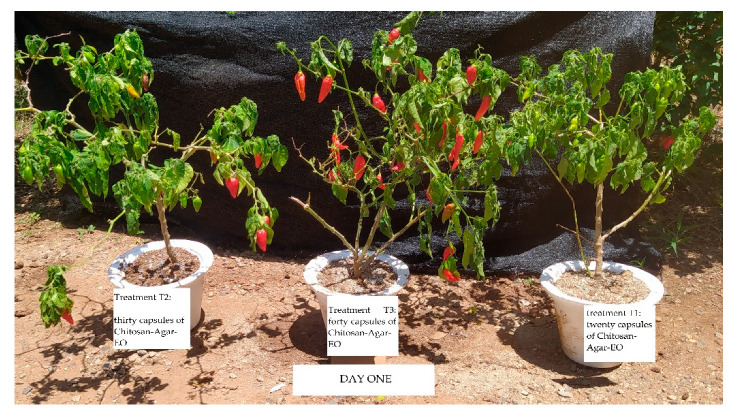
Day 1 of the test to evaluate the antifungal activity of EO capsules in greenhouse conditions: Topito chili plants infected with 1 × 10^6^ conidia mL^−1^ with treatments of 20, 30, and 40 capsules of (Chitosan-Agar-EO) from *Cymbopogon citratus.*

**Figure 8 molecules-25-04447-f008:**
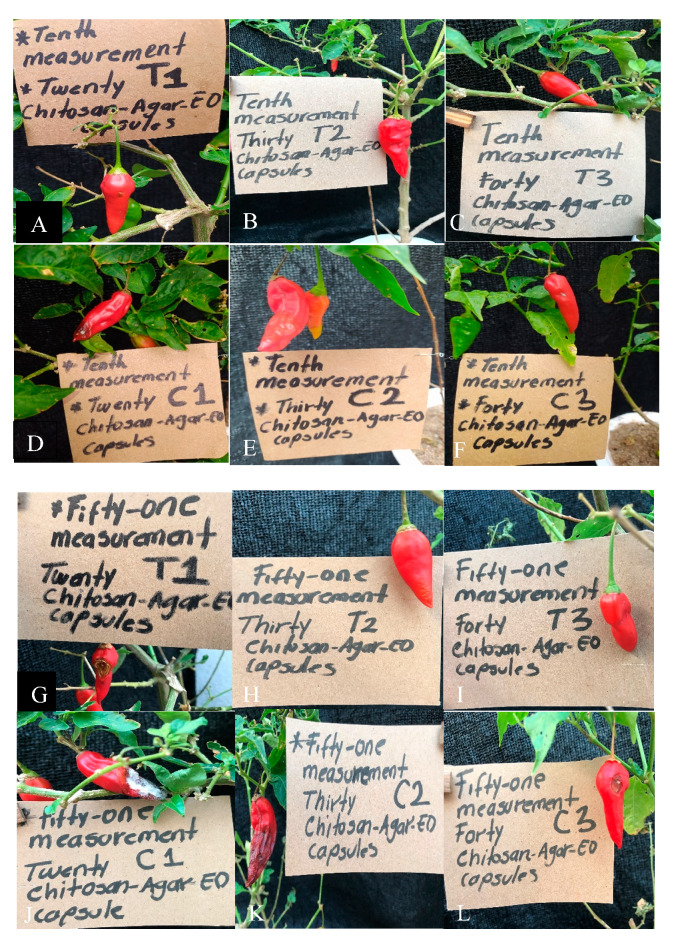
Damage assessment of Topito peppers at different durations. Day 10 (**A**–**F**) and 51 (**G**–**L**) of evaluation: peppers and leaves with slight signs of infection in treatment with 20 capsules of EO and without signs of infection in treatments with 30 and 40 capsules of EO. Peppers and leaves with serious signs of infection in controls with 20, 30, and 40 capsules of corn oil.

**Figure 9 molecules-25-04447-f009:**
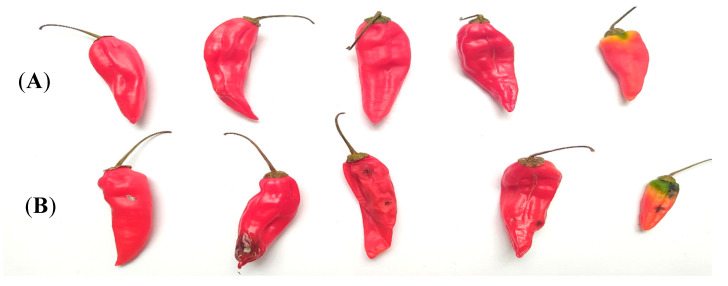
Difference between Topito peppers from infected plants and those treated with capsules (Chitosan-Agar-EO) of *Cymbopogon citratus* (**A**) and Chitosan-AGAR-Corn Oil (**B**).

**Figure 10 molecules-25-04447-f010:**
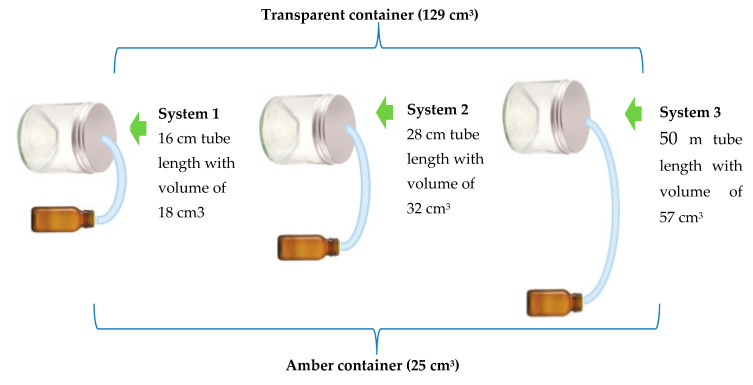
Diagram of the system used for testing the in vitro volatile effect of the de EO-agar-chitosan capsules.

**Table 1 molecules-25-04447-t001:** Ratio of essential oil components of *Cymbopogon citratus.*

Identification	Relative Quantity (%)
6-methyl-5-hepten-2-one	1.2
Myrcene	14.0
α-felandrene	0.1
(*Z*)-β-ocimene	0.6
N.I.	1.4
3-4-methyl-3-pentyl)-furan	0.3
Linalool	2.8
N.I	1.6
N.I	1.3
*cis*-verbenol	3.4
*trans*-verbenol	4.1
N.I.	0.7
Citronellol	1.3
Neral	22.1
3,7-dimethyl-2,6-octadien-1-ol	2.6
Geranial	32.9
2-undecanone	0.7
Eugenol	0.9
geraniol acetate	1.5
(*E*)-Caryophylene	1.7
α-zingibirene	0.4
β-sesquifelandrene	1.6
Tumeron	1.5
Curlona	0.9
mono-2-ethylhexylphthalate	1.2

**Table 2 molecules-25-04447-t002:** Inhibition of *C. gloeosporioides* using different techniques up to 30 days.

Treatment	Inhibition of Mycelial Growth			
	Day 5	Day 10	Day 15	Day 30
Capsules with corn oil (control)	0 ^a^	0 ^a^	0 ^a^	0 ^a^
Poisoned food	88.5 ^c^	68.6 ^c^	30.6 ^b^	0 ^a^
Wet chamber	77.5 ^b^	61.2 ^b^	49 ^c^	0 ^a^
Chitosan-Agar-EO capsules	100 ^d^	100 ^e^	100 ^e^	100 ^b^
*Trichoderma*	88.9 ^c^	90.9 ^d^	96.3 ^e^	98.8 ^b^

Means of the inhibition of mycelial growth values followed by the same superscript letter, vertically, does not differ significantly according to Tukey’s LSD test (*p* ˂ 0.05).

**Table 3 molecules-25-04447-t003:** Antifungal activity of *Cymbopogon citratus* (Chitosan-Agar-EO) capsules against *Colletotrichum gloeosporioides*: percentage of damage in *Capsicum annuum* measured 51 days after inoculation.

Percentage and Degree of Severity of Anthracnose in Topito Peppers with Treatments of EO Capsules of *C. citratus* and Corn Oil Controls				
Capsules of Chitosan-Agar-EO *C. citratus*	**Treatments**	**T1**	**T2**	**T3**
	Average number of peppers with signs of infection	12	0	0
	Number of healthy peppers	25	35	36
	Average number of peppers per plant	37	35	36
Total percentage of damage in each treatment		32% ^ab^	0% ^a^	0% ^a^
Degree of severity		5	0	0
Capsules of Chitosan-Agar-Corn Oil	**Controls**	**C1**	**C2**	**C3**
	Average number of peppers with signs of infection	33	37	35
	Number of healthy peppers	6	4	2
	Average number of peppers per plant	39	41	37
Total percentage of damage in each control		84.6% ^b^	90.2% ^b^	94.6% ^b^
Degree of severity		7	8	9

Different letters as superscript indicate significant differences between treatments according to the Tukey’s honest significance test (*p* ˂ 0.05).

**Table 4 molecules-25-04447-t004:** Treatments and controls used to test the antifungal activity of (Chitosan-Agar-EO) capsules of *Cymbopogon citratus, Poaceae.*

Type of Capsules	Number of Capsules (µL Oil)	Treatments and Controls Utilized		
		System 1 (186 cc)	System 2 (161 cc)	System 3 (147 cc)
(Chitosan-Agar-EO) *C. citratus* (treatments)	20 (170 µL)	(913 ppm)	(1056 ppm)	(1156 ppm)
	30 (255 µL)	(1370 ppm)	(1584 ppm)	(1734 ppm)
	40 (340 µL)	(1827 ppm)	(2111 ppm)	(2312 ppm)
Chitosan-Agar-Corn oil (control)	20 (170 µL)	(913 ppm)	(1056 ppm)	(1156 ppm)
	30 (255 µL)	(1370 ppm)	(1584 ppm)	(1734 ppm)
	40 (340 µL)	(1827 ppm)	(2111 ppm)	(2312 ppm)

**Table 5 molecules-25-04447-t005:** Environmental conditions of the department of Cesar.

	Air Temperature at a Height of 12 m		Precipitation (mm)		Period of Sunlight (hours)		Relative Humidity (%)	
	Month	Average (°C)	Month	Average	Month	Average	Month	Average
Highest	March	35.8	October	199	June	12.7	May	69
	July	36.1			July	12.7	October	75
Lowest	May	32	February	10	December	11.5	January	59
	October	23					February	55
							March	56

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
