# Peer review of "Effectiveness of Cymbopogon citratus Oil Encapsulated in Chitosan on Colletotrichum gloeosporioides Isolated from Capsicum annuum"

_molecules, 2020, doi:10.3390/molecules25194447_

Round 1
Reviewer 1 Report
Following are some of the suggestions for improvement:
Page 1
Line 25: Chitosan capsules present a…
Line 32: products face restrictions…
Line 35: generates causes losses…
Line 37: small farmers…
Page 2
Line 55: are desperately needed…
Page 5
Line 131: During the tests for resistance against environmental conditions, there were no weight losses in capsules for the…
Line 133: After 45 days, the weight…
Page 6
Line 147: 0.0035 g
Line 151-152: While testing the antifungal activity of EO capsules in greenhouse conditions (Fig. 5),…
Page 7
Line 160: Chitosan-Agar-EO from C. citratus.
Line 162: topito pepper fruits at different durations.
Page 8
Line 175: Table 3.
Line 176-177: Please use italics where necessary.
Page 9
Line 184: and 94.6%, respectively.
Line 190: in Capsicum annum, it presents...
Line 202: among other factors. Researchers [6,17] evaluated…
Line 217: associated with several…
Line 217: The first is the…
Page 11:
Line 299: starting the treatment.
Line 312: byproduct obtained from the fishing industry.
Line 324: could be made.
Page 13
Line 394: The solution (85 µl) was…
Line 410: Tubing lengths of 16, 28, and 50 cm… (Please avoid starting a sentence with a numeral)
Page 14
Table 4: Please be uniform with the representation (Chitosan-agar-EO), throughout the text
Line 447-448: de ??? Terminate the statement with one “period” sign.
Line 452: (20, 30, and 40 capsules, respectively)
Page 15
Line 170: 1 ml (please be uniform with the use of units)
Line 486: Please be consistent (See comment for Table 4 above)
Page 16
Line 516: of severity, a degree…
Line 522: than
Author Response
Response to Reviewer 1 Comments
Point 1. Line 25: Chitosan capsules present a…
Response 1: the suggested modification was applied in the text
Point 2. Line 32: products face restrictions…
Response 2: the suggested modification was applied in the text
Ponint 3. Line 35: generates causes losses…
Response 3: the suggested modification was applied in the text
Point 4. Line 37: small farmers…
Response 4: the suggested modification was applied in the text
Point 5. Line 55: are desperately needed…
Response 5: the suggested modification was applied in the text
Point 6. Line 131: During the tests for resistance against environmental conditions, there were no weight losses in capsules for the…
Response 6: the suggested modification was applied in the text
Point 7. Line 133: After 45 days, the weight…
Response 7: the suggested modification was applied in the text
Point 8. Line 147: 0.0035 g
Response 8: the suggested modification was applied in the text
Point 9. Line 151-152: While testing the antifungal activity of EO capsules in greenhouse conditions (Fig. 5),…
Response 9: the suggested modification was applied in the text
Point 10. Line 160: Chitosan-Agar-EO from C. citratus.
Response 10: the suggested modification was applied in the text
Point 11. Line 162: topito pepper fruits at different durations.
Response 11: the suggested modification was applied in the text
Point 12. Line 175: Table 3.
Response 12: the suggested modification was applied in the text
Point 13. Line 176-177: Please use italics where necessary.
Response 13: the suggested modification was applied in the text
Point 14. Line 184: and 94.6%, respectively.
Response 14: the suggested modification was applied in the text
Point 15. Line 190: in Capsicum annum, it presents...
Response 15: the suggested modification was applied in the text
Point 16. Line 202: among other factors. Researchers [6,17] evaluated…
Response 16: the suggested modification was applied in the text
Point 17. Line 217: associated with several…
Response 17: the suggested modification was applied in the text
Point 18. Line 217: The first is the…
Response 18: the suggested modification was applied in the text
Point 19. Line 299: starting the treatment.
Response 19: the suggested modification was applied in the text
Point 20. Line 312: byproduct obtained from the fishing industry.
Response 20: the suggested modification was applied in the text
Point 21. Line 324: could be made.
Response 21: the suggested modification was applied in the text
Point 22. Line 394: The solution (85 µl) was…
Response 22: the suggested modification was applied in the text
Point 23. Line 410: Tubing lengths of 16, 28, and 50 cm… (Please avoid starting a sentence with a numeral)
Response 23: the suggested modification was applied in the text
Point 24. Table 4: Please be uniform with the representation (Chitosan-agar-EO), throughout the text
Response 24: the suggested modification was applied in the text
Point 25. Line 447-448: de ??? Terminate the statement with one “period” sign.
Response 25: the suggested modification was applied in the text
Point 26. Line 452: (20, 30, and 40 capsules, respectively)
Response 26: the suggested modification was applied in the text
Point 27. Line 170: 1 ml (please be uniform with the use of units)
Response 27: the suggested modification was applied in the text
Point 28. Line 486: Please be consistent (See comment for Table 4 above)
Response 28: the suggested modification was applied in the text
Point 29. Line 516: of severity, a degree…
Response 29: the suggested modification was applied in the text
Point 30. Line 522: than
Response 30: the suggested modification was applied in the text
Reviewer 2 Report
I have carefully read manuscript molecules-935814 entitled „Effectiveness of Cymbopogon citratus Oil Encapsulated in Chitosan on Colletotrichum gloeosporioides Isolated from Capsicum annuum“.Extraction and phytochemical analysis of EO C. citratus together with encapsulation in chitosan agar were performed to compare with other techniques and determine its effect on C. gloeosporioides. The capsules showed an effective period of 51 days, with an additional 30 days of efficacy after the reinfection cycle; providing similar results as in the control of Trichoderma sp. Chitosan capsules propose a promising strategy for the use of C. citratus EO on C. gloeosporioides, which are highly effective and stable in vitro and in the field.
The scope of the paper is of interest and I have found a general good quality of the research. From experimentation to data evaluation, everything is well organized and clearly described and the GC-MS analysis and MIC experiments appears to be carefully performed. In my opinion, this work could be accepted to be published in Molecules, after minor revision. All my remarks are summarized as follows:
Line 84-91 Please, provide GC/MS chromatogram of investigated sample.
Supplementary You must translate everything that is written as well as the titles of the documents into English.
Author Response
Response to Reviewer 2 Comments
Point 1: Line 84-91. Please, provide GC/MS chromatogram of investigated sample.
Response 1:
We included the chromatogram of the essential oil of C. citratus, but we propose that the table be kept in the text as a complement.
Point 2: Supplementary You must translate everything that is written as well as the titles of the documents into English.
Response 2:
Suggestion accepted and all headings translated
Reviewer 3 Report
The manuscript ID: molecules-935814- title: Effectiveness of Cymbopogon citratus Oil Encapsulated in Chitosan on Colletotrichum gloeosporioides Isolated from Capsicum annuum- describes the effectivity of the chitosan-agar capsules containing EO, from Cymbopogon citratus, with respect EO for the control of C. gloeosporioides isolated from sweet pepper plants and propose the capsules as potential biopesticide.
The manuscript needs major revision particularly on chitosan-agar capsules characterization.
Section 2. Materials and methods need to be improved.
Authors should provide more information on the characteristics of the chitosan-agar capsules containing EO such as chitosan-agar capsules preparation (section 2.3). Moreover, the physical and morphological characterization of chitosan-agar capsules, the EO encapsulation efficiency and the in vitro release of the EO are absent.
The manuscript needs the addition of appropriate controls in the experimental section on the fungus.
Author Response
Response to Reviewer 3 Comments
Point 1: Authors should provide more information on the characteristics of the chitosan-agar capsules containing EO such as chitosan-agar capsules preparation (section 2.3). Moreover, the physical and morphological characterization of chitosan-agar capsules, the EO encapsulation efficiency and the in vitro release of the EO are absent.
Response 1:
The reviewer's suggestions were accepted and the requested adjustments which included an expanded description of the preparation of the capsules as well as the morphological characterization of these were added to the document in the methodology and results sections. The methodologies used to calculate the encapsulation efficiency described by (AOAC, 2005), Vila (2001) and (Calvo et al., 2010) were also included. Finally, the calculation of in vitro release was presented according to (Mishra et al., 2015).
Point 2: The manuscript needs the addition of appropriate controls in the experimental section on the fungus
Response 2:
Chemical controls for the management of the fungus were not included since the objective of the study was to present an ecological alternative for its management. In order to maintain the ecological orientation of the study, a comparison of the efficiency of the control of the pathogen represented in a control was performed on a raw material whose active ingredient is Trichoderma sp.
Round 2
Reviewer 3 Report
no comments
Author Response
the text was revised again by a native speaker expert translator.
